# Intermittent Pneumatic Compression and Cold Water Immersion Effects on Physiological and Perceptual Recovery during Multi-Sports International Championship

**DOI:** 10.3390/jfmk5030045

**Published:** 2020-06-30

**Authors:** Ismael Martínez-Guardado, Daniel Rojas-Valverde, Randall Gutiérrez-Vargas, Alexis Ugalde Ramírez, Juan Carlos Gutiérrez-Vargas, Braulio Sánchez-Ureña

**Affiliations:** 1Grupo de Avances en Entrenamiento Deportivo y Acondicionamiento Físico (GAEDAF), Faculty of Sports Science, University of Extremadura, 10005 Cáceres, Spain; 2Centro de Investigación y Diagnóstico en Salud y Deporte (CIDISAD), Escuela Ciencias del Movimiento Humano y Calidad de Vida (CIEMHCAVI), Universidad Nacional, Heredia 86-3000, Costa Rica; randall.gutierrez.vargas@una.cr (R.G.-V.); augalde07@hotmail.com (A.U.R.); 3Centro de Estudios para el Desarrollo, Rehabilitación y Salud (CEDERSA), Escuela Ciencias del Movimiento Humano y Calidad de Vida (CIEMHCAVI), Universidad Nacional, Heredia 86-3000, Costa Rica; jucagu@msn.com; 4Programa de Ciencias del Ejercicio y la Salud (PROCESA), Escuela Ciencias del Movimiento Humano y Calidad de Vida (CIEMHCAVI), Universidad Nacional, Heredia 86-3000, Costa Rica

**Keywords:** hydration, pain, tensiomyography, muscle function, team sports

## Abstract

Background: Congested-fixture championships are common during the selection of the athletes and teams participating in the Olympic Games. Throughout these tournaments, it is fundamental to perform optimally, rest well, and recover between competitions. This study aimed to (a) explore the effectiveness of the use of intermittent pneumatic compression (IPC) and cold water immersion (CWI) to recover muscle mechanical function (MuscleMechFx), hydration status (HydS), pain perception (PainPercep), rate of perceived exertion (RPE), sleep hours, and sleep quality (SleepQual) during a regional multi-sports international championship and (b) compare these results by sex. Methods: A total of 52 basketball and handball players were exposed to a recovery protocol after the competition as follows: IPC, sequential 20 min at 200 mmHg, and CWI, continuous 12 min at 12 °C. Results: MuscleMechFx presented differences by match and sex (*p* = 0.058) in time of contraction of biceps femoris; SleepQual and sleep hours were different between matches (<0.01). Conclusions: IPC + CWI seems to be effective to maintain some MuscleMechFx, HydS, and recovery and pain perception during a congested multi-sport tournament.

## 1. Introduction

In recent years, sports dynamics have changed due to the increased frequency and intensity of competition. As a consequence, team sports have become more competitive, with an increasingly condensed game schedule [1]. In this line, during the congested fixture period, team sports players are often required to play competitive matches with one–three days of recovery in between [2,3]. This is why constant assessment of the different physical and physiological parameters during these periods is necessary to maintain the optimal functional and structural integrity of the athlete’s body [2]. Team sports, in this case, basketball and handball, are called intermittent effort sports due to many actions performed at high and low intensity interspersed with each other [4,5]. Further, due to the high frequency with which team sports athletes train and compete, recovery methods are essential in sports performance [6].

Recovery is defined as a multi-faceted restorative process related to time, both physical and physiological [7]. However, it has been previously shown that these types of sports also cause mental fatigue which must be treated through mental recovery [8]. Fatigue is considered as a state of increased tiredness produced by physical and mental exertion, and since it is considered as a reduction in physical or functional performance, its control is essential concerning sports performance [9]. In this line, the presence of fatigue and long-term insufficient recovery leads to the unfavorable development of the performance of athletes, and the structural disruptions that it generates can be maintained for a period of between three and seven days [10]. Thus, in the congested fixture periods, it is difficult to maintain the optimal status of the athletes due to being exposed to less than 72 h of recovery between games and training sessions before fully recovering from additional training loads and with a higher accumulation of fatigue [11,12].

There are several scientifically proven and frequently used post-exercise recovery methods for athletes, such as cold water immersions (CWI), compressions garments, sleep, stretching, swimming, walking, or jogging [13,14,15]. Further, hydration and nutrition have been proven to be important during the recovery process [7,12]. According to cold water immersion, several studies have reported the beneficial effect of these methods in recovery [16,17,18], and specifically in generating reductions in acute muscle inflammation, muscle spasms and sensations of pain, and symptoms related to delayed onset muscle soreness (DOMS), among others [19,20,21]. In this line, this type of recovery method could be a practical tool to be included during a congested fixture period [22]. However, Moreno et al. [23] observed the importance of individuality in recovery since Spanish basketball players used different recovery methods and reported different perceptions about them. On the other hand, compression garments have been used for exercise-related fatigue recovery. One of these methods is the intermittent pneumatic compression (IPC) pump, which consists of gradual pressure (0–300 mmHg) gradients applied to facilitate lymph and blood flow [24]. In sports, it has been shown that IPC systems may allow the retained fluid and substances (e.g., metabolites) related to muscle damage and neuromuscular fatigue to be removed from the lower and upper limbs [25]. This recovery method is generally designed to simulate the muscle pump, like a typical massage.

Furthermore, it is increasingly common for athletes and their respective medical teams to recognize the importance of adequate sleep duration and quality in sports performance and recovery [26]. As sleep researchers and clinicians learn more about the effects of sleep duration and quality on human performance and recovery, it is understandable that athletes begin to apply these principles to take advantage of their competitors. However, sleep deprivation can put athletes at risk of injury, and therefore plays an important role in the recovery of these athletes [27]. Thus, it has been previously established that there is a relationship between sleep duration and competition performance [28]. However, even if athletes have an optimal sleep duration, they are likely to have primary sleep disorders that can affect sleep quality, and these sleep disorders often go unnoticed [29]. In this line, previous studies have correlated poor sleep quality and loss in a competition [30]. Moreover, Luke et al. [31] reported that sleeping less than 6 h is associated with the presence of injuries in young athletes. However, it has been reported that athletes who sleep more than 8 h are more likely to be injured as well [32]. For this reason, teams must consider the optimal dose so that their athletes have an adequate duration and quality of sleep.

This study aimed to a) explore the effectiveness of the use of intermittent pneumatic compression (IPC) and cold water immersion (CWI) to recover muscle mechanical function (MuscleMechFx), hydration status (HydS), pain perception (PainPercep), rate of perceived exertion (RPE), and sleep quality (SleepQual) during a regional multi-sports international championship and b) compare these results by sex.

## 2. Materials and Methods

### 2.1. Experimental Design

Players were exposed to a recovery protocol (20 min IPC + 12 min CWI) after each match during the regular phase of a congested fixture tournament. Participants were assessed in three areas: hydration status (urine specific gravity, body weight, and urine solids), muscle mechanical function of biceps femoris and rectus femoris (muscle radial deformation [Dm] and time of contraction [Tc]), and perceived recovery and pain (total quality recovery, delayed onset muscle soreness, and sleep hours and quality). Figure 1 shows the chronological assessment of the variables. Muscle mechanical function was assessed pre-matches, and ∼1 h after each match, hydration status, perceived recovery, and pain were assessed pre-match and the morning after each match (∼6–7 a.m.). The rate of perceived exertion was registered during matches to have an internal control of the effort.

### 2.2. Participants

Fifty-two athletes of two different disciplines took part in the study. There were 26 women (age 23.1 ± 8.9 years, height 168.2 ± 8.4 cm, and weight 66.01 ± 8.68 kg) (14 handball and 12 basketball players) and 26 men (age 24.5 ± 9.2 years, height 175.2 ± 9.7 cm, and weight 84.68 ± 17.48 kg) (14 handball and 12 basketball players). All athletes were members of a national team participating in the Managua Central American Games, a regional multi-sport championship, which was part of the road to the 2020 Olympic Games. There were no reported adverse neuromuscular or metabolic reports at least 3 months before the competition.

All participants gave their written informed consent to participate in the study protocol. They were informed of the details of the procedures and the associated potential risk or discomforts as well as their benefits and rights. The protocol followed the criteria of the Declaration of Helsinki regarding biomedical research in humans (18th Medical Assembly, revised in Fortaleza, 2013) and it was previously approved by a review board of National University, Costa Rica (CEC-CON; Code= 2016-17).

### 2.3. Instruments and Procedures

#### 2.3.1. Muscle Mechanical Function

Muscle radial deformation (Dm, mm) and time of contraction (Tc, ms) of the reported dominant rectus femoris (RF) and biceps femoris (BF) were assessed using a tensiomyography system (TMG, Lubljana, Slovenia) and followed previously reported protocols [2,33,34]. These two parameters have had good reliability (Tc, ICC = 0.92 and Dm, ICC = 0.94–0.97) [35]. Participants were asked to remain relaxed for 5 min in the supine position and with the knee fixed at 5° (BF) and 120° (RF) using cushioned pads. After cleaning the area, two 5 cm^2^ adhesive electrodes (TheraTrode^®^, TheraSigma, Washougal, Washington, USA) were placed in the muscle at a distance of 5 cm from each other and avoiding tendon insertions. The measurement point was set at the maximum radial circumference of each muscle. These points were selected visually and by palpation during a voluntary contraction. A digital displacement transducer (GK 40, Panoptik doo, Ljubljana, Slovenia) was placed perpendicular to this selected point. Electrodes were connected to an electrical stimulator (TMG-S2 doo, Ljubljana, Slovenia) and 40 mA single-phase wave stimuli were triggered to induce muscle contraction, increasing by 20 mA until the maximum radial displacement was obtained. The electrical stimuli were separated from each other by 10 s of rest, to avoid fatigue or post-tetanic activation [36].

#### 2.3.2. Perceived Recovery and Pain

Three variables were measured to assess perceived recovery: reported perceived perception, sleep quality, and muscle pain. Total quality recovery (TQR) was assessed by using a visual analog scale [37] between 6 and 20 in which players pointed out their level of recovery perceived, 6 being “no recovery at all”, and 20 being “maximal recovery”. Delayed onset muscle soreness (DOMS) was measured by using a visual scale of pain where the athletes would choose between 0 and 10 the level of pain they were perceiving after executing a 2 s 90° squat, 0 meaning “no pain” and 10 meaning “extreme pain” [38]. Sleep quality was assessed using a visual analog scale, 0 meaning the worst possible sleep and 11 the best possible sleep [39]. Players self-reported sleeping hours.

To evaluate the effort made in the matches, the rate of perceived exertion (RPE) was asked to the players immediately after each match. It was measured with the Borg Scale 6–20 [40], where 6 was interpreted as an effort “very, very light” and 20 as a “Maximum, strenuous” effort.

#### 2.3.3. Hydration Status

Hydration was assessed through body weight (BW, kg), urine specific gravity (USG), and urine solids (USol). BW was measured using a digital scale (Tanita, Ironman, CA, USA). USG and USol were measured using a valid [41] digital handheld refractometer (Palm Abbe^TM^, Misco, OH, USA) and classified following previously established ranges: well-hydrated <1.01, minimal dehydration 1.01–1.02, significant dehydration 1.02–1.03, and serious dehydration >1.03 [42]. The refractometer was cleaned with distilled water and calibrated previously. There were no reported urination difficulties.

#### 2.3.4. Post-Competition Recovery Protocols

The recovery protocol consisted of the combination of intermittent pneumatic compression (IPC) and cold water immersion (CWI). The IPC + CWI protocol was applied ~45 min after each match. IPC consisted of a sequential (circulation mode) 20 min [25,43,44] compression at 200 mmHg [24]. The air compression system (LX7 [E0651], Doctor Life^®^ Healthcare, Seoul, Korea) had 4 chambers (foot, calf, knee, thigh) that applied intermittent compression from foot to groin in both lower limbs independently in cycles of 60 s (inflation 40 s [10 s per chamber] and pause time 20 s) [45]. Participants were in the supine and rest position during the IPC protocol. After the IPC, a continuous CWI was applied using portable pools (iSquad, Gold Coast, Australia) and automated electric cooling systems (iCool Compact, Gold Coast, Australia). The participants remained seated and water was set at chest level. CWI was performed for 12 min at a temperature of 12 °C [20,33].

### 2.4. Statistical Analysis

Data was presented as mean and standard deviation (m ± sd). Data normality was confirmed using the Kolmogorov–Smirnov test. Mixed analysis of variance was performed to compare measures vs. sex in all variables, and post hoc analysis was performed using the Bonferroni method. Magnitudes of differences for all variables were analyzed using the partial omega squared (*ω_p_^2^*) and qualitatively interpreted using the following thresholds: <0.01 trivial, >0.01 small; >0.06 medium, and >0.14 large [46]. Alpha was set at *p* < 0.05. The data analysis was performed using Statistical Package for the Social Sciences (IBM, SPSS Statistics, V 22.0 Chicago, IL, USA).

## 3. Results

### 3.1. Muscle Mechanical Function

Table 1 presents the differences in muscle mechanical function between matches by sex. There were found differences by match in BF Tc (*p* = 0.005, second match > pre matches; *p* = 0.002, third match > pre matches), RF Tc (*p* = 0.023; second match < pre matches), and BF Dm (*p* = 0.007, second match > first match; *p* = 0.02, third match > pre matches), and by sex in BF Tc (*p* = 0.002, women > men) and Dm (*p* = 0.024, women > men).

### 3.2. Perceived Recovery

There were significant differences in sleep quality by match (*p* = 0.002, third match < pre matches; *p* = 0.02, second match < pre matches and *p* < 0.01, third match < second match) (see Table 2) and in sleep hours (*p* = 0.003, third match < pre matches; *p* = 0.016, third match < first match).

### 3.3. Hydration Status

There were differences in body weight between matches (see Table 3), but in response to the differences by sex found (*p* = 0.001, women < men), this was confirmed by no changes in USG or USol during the championship throughout the matches by neither sex.

## 4. Discussion

To the best of our knowledge, this is the first study that investigated the effectiveness of the combination of IPC and CWI to recover muscle mechanical function, hydration status, pain perception, rate of perceived exertion, and sleep quality during a regional multi-sports international championship and compared these results by sex. The most interesting finding was that Tc and Dm were increased in the biceps femoris after each match and women presented higher values. Further, sleep quality and sleep hours changed after each match. Body weight was higher after each match in men.

Regarding muscle mechanical function, TMG is a non-invasive method that allows the assessment of contractile muscle properties under isolation conditions without producing additional fatigue [36,47]. In the present study, TMG-related fatigue parameters (Dm and Tc) were measured. In this line, Dm is being considered as a measure of muscle stiffness and Tc is associated with muscle fatigue rate [48]. Our results showed that the implementation of an IPC and CWI recovery protocol provoked the main effect of match and sex in Tc and Dm in BF and a slight main effect of match in Tc in RF. As previous studies have been established, the increments in Tc and Dm refer to an overall deterioration in the neural response and a loss in muscle stiffness, respectively [34,49]. In this line, other studies have shown that immersions in cold water do not contribute significantly to the recovery of the muscular function in university active students [33]. Comparing both sexes, it has been demonstrated that human skeletal muscle fatigue is influenced by the biological sex of the individual [50]. Further, the sex differences in Tc and Dm in BF reported in the present study could be attributed to the morphology and body composition characteristics of each sex. In this sense, these differences could be attributed to the segment length and muscle mass in men [51] presenting higher muscle fatigue parameters than women.

Concerning TQR and DOMS, no differences were found between groups on these parameters. Despite the lack of differences, the recovery protocol applied in the present study caused that both sexes had a high value of TQR and lower values of DOMS. It seems that several perceptual parameters, such as stress, quality of sleep, fatigue, and DOMS, among others, can be affected by the level of the load imposed by training sessions and matches [52]. Moreover, previous studies that have applied CWI after a competition match have shown its effectiveness to reduce physiological and functional signs related to DOMS [20,21,53]. Moreover, a combined CWI + compression recovery method was effective to reduce, in an acute and prolonged way, perceived soreness in professional male tennis players [54]. Besides, other studies have shown lower levels of DOMS and fatigue up to three days before a competitive match during a congested fixture period compared with the week of preparation for a match in the regular competitive phase in elite handball players. In this line, congested fixture periods involve a significantly lower training load before the match than regular weeks, which could help to explain this fact. Moreover, RPE also did not differ between the two groups evaluated. Thus, despite training loads during congested fixture periods that are reduced, other studies have reported that this reduction was not enough to prevent disruption of the players’ wellbeing [55]. Similar to our results, Sánchez-Ureña et al. [33] reported similar RPE values after two CWI protocols compared to a control group. Further, Duffield et al. [54] showed similar responses in RPE between recovery (CWI + compression methods) and a control group (no recovery).

Concerning sleep variables, a main effect on the match was observed. In both quality and quantity, women reported higher values than men before starting the congested fixture period and during it. As has been previously reported, the amount of sleep appears to have a large impact on sports performance [18]. Cold exposure causes changes in the neurotransmitters dopamine and serotonin which can affect sleep [20]. Besides, CWI can rapidly lower core and skin temperatures after exercise and maintain it 90 min post-immersion [56]. Thus, post-exercise CWI recovery methods could accelerate the decline of core temperature and therefore augment sleep propensity [57]. According to sex differences, Koikawa et al. [58] reported that collegiate female soccer players presented poorer subjective sleep quality than male players. However, the authors established that the underpinning mechanism of this fact is unclear.

During exercise, although not so easy to observe, water loss (dehydration) rapidly reduces performance in training and match competition [12]. Thus, athletes must have an adequate level of hydration, especially in congested fixture periods where the recovery time between games is very short. In this line, techniques that sample body fluids in real-time and are easy to use (i.e., urine specific gravity and urine osmolality) provide good agreement regarding hydration status during in-field sampling [59]. However, in the present study, no differences in urine specific gravity and urine solids were noticed. Despite this, body weight could be considered a good indicator of fluid balance during competition. Our results show differences in body weight with a loss after each match and also differences by sex. This should be taken into account by coaches, medical staff, and athletes to monitor their hydration status and food intake during the competition to better perform.

## 5. Limitations

While this study aimed to explore the effectiveness of the use of intermittent pneumatic compression (IPC) and cold water immersion (CWI) to recover muscle mechanical function (MuscleMechFx), hydration status (HydS), pain perception (PainPercep), rate of perceived exertion (RPE), and sleep quality (SleepQual) during a regional multi-sports international championship and compared these results by sex, some limitations should be acknowledged. First, the natural origin of these types of tournaments did not allow the researchers to have a control group to be able to compare the effectiveness of the protocols, therefore explaining the descriptive nature of the present study. This organic condition also limited the possibility of using technological tools such as heart rate monitoring or external load using local or global location systems for the quantification of the competition load, and only the recording of the rate of perceived exertion was assessed.

## 6. Conclusions

In conclusion, the data presented herein indicate that during a very short congested fixture period, composed of three handball and basketball matches in one week, a combined recovery method (IPC + CWI) could maintain muscle mechanical properties, perceived recovery, and hydration status throughout the tournament, but there are some differences by sex that should be addressed by coaches, medical staff, and sports scientists to optimize prescribed recovery strategies for women and men.

## Figures and Tables

**Figure 1 jfmk-05-00045-f001:**
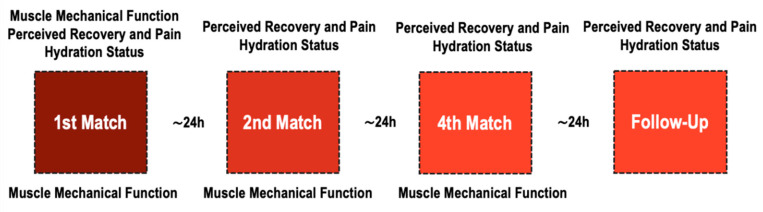
Chronological assessment of variables throughout a multi-sport championship.

**Table 1 jfmk-05-00045-t001:** Analysis of muscle mechanical function throughout matches by sex.

Muscle	Parameter	Sex	Pre Matches	1st Match	2nd Match	3rd Match	F match (*p*-Value)	ω_p_^2^ (Rating)
Biceps Femoris	Tc (ms)	Male	31.3 ± 10.03	32.91 ± 12.79	34.96 ± 13.53	35.64 ± 15.21	5–14 (0.005)	0.25 (Large)
		Female	35.46 ± 11.33	48.79 ± 17.51	43.87 ± 16.24	53.18 ± 17.49		
		F sex (*p*-value)	11.13 (0.002)				F Interaction (*p*-value)	
		ω_p_^2^ (rating)	0.46 (Large)				2.58 (0.058)	
	Dm (mm)	Male	4.6 ± 2.63	4.73 ± 2.73	5.34 ± 2.72	5.63 ± 3.09	3.5 (0.03)	0.17 (Large)
		Female	5.81 ± 1.92	7.21 ± 2.35	7.42 ± 2.45	7.28 ± 2.35		
		F sex (*p*-value)	5.61 (0.024)				F Interaction (*p*-value)	
		ω_p_^2^ (rating)	0.28 (Large)				1.1 (0.35)	
Rectus Femoris	Tc (ms)	Male	28.65 ± 5.53	27.11 ± 4.95	26.61 ± 5.06	27.36 ± 5.69	3.23 (0.04)	0.16 (Large)
		Female	28.17 ± 3.21	27.27 ± 4.09	26.08 ± 3.03	26.52 ± 2.12		
		F sex (*p*-value)	0.1 (0.77)				F Interaction (*p*-value)	
		ω_p_^2^ (rating)	0.08 (Small)				0.19 (0.9)	
	Dm (mm)	Male	6.26 ± 1.72	6.85 ± 2.16	6.81 ± 2.24	7.13 ± 2.19	2.05 (0.13)	0.08 (Medium)
		Female	7.08 ± 2.31	8.06 ± 2.16	7.93 ± 1.97	7.84 ± 1.95		
		F sex (*p*-value)	2.79 (0.104)				F Interaction (*p*-value)	
		ω_p_^2^ (rating)	0.13 (Medium)				0.23 (0.88)	

**Table 2 jfmk-05-00045-t002:** Perceptual recovery comparison throughout matches by sex.

Parameter	Sex	Pre Matches	1st Match	2nd Match	3rd Match	F match (*p*-Value)	ω_p_^2^ (Rating)
Total Quality Recovery	Male	16.99 ± 1.57	17.5 ± 1.16	18.21 ± 1.25	17.21 ± 1.19	2.47 (0.087)	0.13 (Medium)
	Female	16.79 ± 2.64	16.79 ± 1.02	16.5 ± 1.69	16.43 ± 1.87		
	F sex (*p*-value)	2.6 (0.12)				F Interaction (*p*-value)	
	ω_p_^2^ (rating)	0.13 (Medium)				1.23 (0.31)	
Delayed Onset Muscle Soreness	Male	0.14 ± 0.53	0.43 ± 1.59	0.39 ± 1.24	0.86 ± 2.21	1.53 (0.233)	0.02 (Small)
	Female	0.71 ± 1.98	0.29 ± 1.07	0.64 ± 1.45	1.21 ± 1.53		
	F sex (*p*-value)	1.095 (0.31)				F Interaction (*p*-value)	
	ω_p_^2^ (rating)	0.01 (Small)				0.493 (0.69)	
Sleep Quality	Male	8.79 ± 0.98	7.29 ± 0.61	8.07 ± 0.83	7.93 ± 1.07	18.91(<0.01)	0.6 (Large)
	Female	7.86 ± 0.54	8.21 ± 0.58	8.64 ± 0.84	7.29 ± 0.47		
	F sex (*p*-value)	0.01 (0.923)				F Interaction (*p*-value)	
	ω_p_^2^ (rating)	0.08 (Small)				12.12 (<0.01)	
Sleep hours	Male	8.79 ± 0.58	8.64 ± 0.75	8.64 ± 1.15	8.64 ± 1.01	5.05 (0.007)	0.25 (Large)
	Female	9.5 ± 0.76	8.99 ± 1.04	9.14 ± 1.03	7.79 ± 1.31		
	F sex (*p*-value)	0.45 (0.51)				F Interaction (*p*-value)	
	ω_p_^2^ (rating)	0.05 (Small)				5.67 (0.001)	
Rate of Perceived Exertion	Male	NA	6.21 ± 1.72	6.49 ± 2.59	6.29 ± 2.23	0.34 (0.72)	0.05 (Small)
	Female	NA	7.78 ± 1.48	6.99 ± 2.54	7.69 ± 1.37		
	F sex (*p*-value)	3.61 (0.07)				F Interaction (*p*-value)	
	ω_p_^2^ (rating)	0.18 (Large)				0.841 (0.44)	

**Table 3 jfmk-05-00045-t003:** Change of hydration status variables between matches by sex.

Parameter	Sex	Pre Matches	1st Match	2nd Match	3rd Match	F match (*p*-Value)	ω_p_^2^ (Rating)
Urine Specific Gravity	Male	1.02 ± 0.01	1.018 ± 0.01	1.016 ± 0.01	1.019 ± 0.01	0.51 (0.682)	0.08 (Small)
	Female	1.015 ± 0.004	1.015 ± 0.01	1.087 ± 0.27	1.016 ± 0.01		
	F sex (*p*-value)	0.729 (0.401)				F Interaction (*p*-value)	
	ω_p_^2^ (rating)	0.08 (Small)				1.08 (0.362)	
Urine Solids	Male	4.82 ± 1.55	4.31 ± 1.4	3.71 ± 1.17	4.37 ± 1.79	2.34 (0.099)	0.1 (Medium)
	Female	3.64 ± 1.24	3.56 ± 1.38	3.53 ± 1.31	3.99 ± 1.42		
	F sex (*p*-value)	2.39 (0.14)				F Interaction (*p*-value)	
	ω_p_^2^ (rating)	0.1 (Medium)				1.137 (0.34)	
Body Weight	Male	84.68 ± 17.48	85.29 ± 17.36	84.87 ± 17.35	84.91 ± 17.27	3.62 (0.028)	0.18 (Large)
	Female	66.01 ± 8.68	65.94 ± 8.89	65.92 ± 8.89	66.08 ± 8.92		
	F sex (*p*-value)	13.24 (0.001)				F Interaction (*p*-value)	
	ω_p_^2^ (rating)	0.5 (Large)				0.01 (0.924)	

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
