# Peer review of "Intermittent Pneumatic Compression and Cold Water Immersion Effects on Physiological and Perceptual Recovery during Multi-Sports International Championship"

_jfmk, 2020, doi:10.3390/jfmk5030045_

Round 1

Reviewer 1 Report

This is a nice study which investigates the use of a combined recovery approach using both compression and cold-water immersion for recovery from high level team sport. It adds to the literature in the area but without a control group or groups which used compression and cold-water immersion independently it’s difficult to draw clear conclusions from this to inform practice. Limitations such as this are not well addressed in the discussion but nor are unfounded claims around the significance of the work made. The paper is very well written, data analysed and presented in an appropriate and clear manner. I have highlighted some specific comments which could be considered and may improve the paper in each section.

Introduction:

A clear introduction which contextualises the need for this work well. A good range of appropriate sources are included and specific issues for intermittent high intensity team sports highlighted.

There is little emphasis or prior literature review around intermittent compression included here though so this comes a little out of the blue in the aim. Consider a little more balance between the two techniques you have chosen and why a combined approach is used here.

Methods:

It would be nice to see a rationale presented for the timings chosen for both IPC and CWI. This doesn’t really come through in your introduction or methods so feels a little arbitrary.

Good to see a gender balance and high level of athletes in your participants.

Without a control group it is difficult to see how you can draw much from this data set to help inform practice.

Heart rate or some GPS/performance type data on intensity may have been useful here.

Results:

A clearly presented data set is included here.

Tables are well laid out and it clear to see the main findings of the study.

Discussion:

This section pulls out key findings and compares with prior research well.

Limitations of the study could come be discussed in more detail in some of the comments made here.

Line 230. A ‘slightly main effect’?? Odd wording.

Conclusion:

A fair conclusion is drawn and linked to professional practice in elite sport.

Author Response

General comments

R1.1. This is a nice study which investigates the use of a combined recovery approach using both compression and cold-water immersion for recovery from high level team sport. It adds to the literature in the area but without a control group or groups which used compression and cold-water immersion independently it’s difficult to draw clear conclusions from this to inform practice. Limitations such as this are not well addressed in the discussion but nor are unfounded claims around the significance of the work made. The paper is very well written, data analysed and presented in an appropriate and clear manner. I have highlighted some specific comments which could be considered and may improve the paper in each section.

We really appreciate your recommendations and opportunity to clarify some issues. Thank you.

R1.2. Introduction:

A clear introduction which contextualises the need for this work well. A good range of appropriate sources are included and specific issues for intermittent high intensity team sports highlighted.

There is little emphasis or prior literature review around intermittent compression included here though so this comes a little out of the blue in the aim. Consider a little more balance between the two techniques you have chosen and why a combined approach is used here.

- Statement: We agree with your comment. We thank the reviewer for this feedback. The following information has been added in the introduction to balance the information of both recovery protocols: “On the other hand, compressions garments have been used for exercise-related fatigue recovery. One of these methods is the intermittent pneumatic compression (IPC) pump, which consists of a gradual pressure (0-300 mmHg) gradients applied to facilitate lymph and blood flow [24]. In sports, it has been shown that IPC systems may allow the retained fluid and substances (eg. metabolites) related to muscle damage and neuromuscular fatigue to be removed from the lower and upper limbs [25]. This recovery method are generally designed to simulate the muscle pump, like a typical massage”.

R1.3. Methods:

It would be nice to see a rationale presented for the timings chosen for both IPC and CWI. This doesn’t really come through in your introduction or methods so feels a little arbitrary.

- Statement: We thank the reviewer for this legitimate comment. The protocols used are similar to those used in previous studies. Appropriate references have been added in this section. This issue was clarified and now the rationale was improved regarding the use of this protocols and specific times.

Good to see a gender balance and high level of athletes in your participants.

- Statement: Thanks for your comment.

Without a control group it is difficult to see how you can draw much from this data set to help inform practice.

- Statement: Unfortunately, we did not have the opportunity to include a control group. This has been added in the limitations section.

Heart rate or some GPS/performance type data on intensity may have been useful here.

- Statement: Unfortunately, we did not have the opportunity to uses these technological tools. This has been added in the limitations section too.

R1.4. Results:

A clearly presented data set is included here. Tables are well laid out and it clear to see the main findings of the study.

- Statement: We thank the reviewer for this comment.

R1.5. Discussion:

This section pulls out key findings and compares with prior research well.

- Statement: We thank the reviewer for this comment.

R1.6. Limitations of the study could come be discussed in more detail in some of the comments made here.

- Statement: We apologize for not providing this information. The limitations section has been added with the following information: “While this study aimed to explore the effectiveness of the use of intermittent pneumatic compression (IPC) and cold water immersion (CWI) to recover muscle mechanical function (MuscleMechFx), hydration status (HydS), pain perception (PainPercep), rate of perceived exertion (RPE) and sleep quality (SleepQual) during a regional multi-sports international championship and b) compared these results by sex; some limitations should be knowledgeable. First, the natural origin of this type of tournaments did not allow the researchers to have a control group to be able to compare the effectiveness of the protocols, therefore the descriptive nature of the present study. This organic condition also limited the possibility of using technological tools such as heart rate monitoring or external load using local or global location systems for the quantification of the competition load and only the recording of the rate of perceived exertion was assessed”.

Line 230. A ‘slightly main effect’?? Odd wording.

- Statement: Thanks for your contribution. This has been changed by “slight main effect”.

R1.7. Conclusion:

A fair conclusion is drawn and linked to professional practice in elite sport.

- Statement: We thank the reviewer for this comment.

Reviewer 2 Report

The authors attempt to investigate a combination of intermittent pneumatic compression and cold water immersion recovery techniques on various physiological and perceptual measures following basketball and handball matches during a busy competition period. The combined recovery techniques appear to have had beneficial effects on recovery, although these benefits depended on the sex of the athlete.

The authors are to be congratulated on their fine study. While this work is promising, there is a major concern with the Introduction of the manuscript and some minor issues with English language and style that will be discussed.

Introduction: Of all of the sections of the manuscript, the Introduction did not adequately address the rationale for the study. In essentially one journal page, the authors attempt to address the importance of recovery and some of the methods for optimizing recovery. Yet, nothing is mentioned about using intermittent pneumatic compression. The authors provide a rationale for cold water immersion, but not for IPC (despite using the two in combination). The authors reference other IPC studies in the Discussion, so there are studies that could be included in the Introduction to bolster the strength of that part of the manuscript. The other sections (i.e. Methods, Results, and Discussion) are nicely presented.

Minor issues: The manuscript evidences throughout some minor issues with language, punctuation, capitalization, etc. What follows is a brief list of examples:

Page 1/Line 19: Capitalize May

Page 1/Line 21: Delete is its and replace with it is 

Page 1/Line 26: Replace compared with compare 

Page 2/Line 44: Change thus to these 

Page 2/Line 46: Delete on 

Page 3/Line 104: Rewrite to read ...took part in the study. There were... 

Page 4/Line 132: Do you mean to use the word form instead of from

Page 4/Line 156: I think the TM should be superscript. [Palm Abbe]

Page 6/Line 210: body weight not body Weight 

Page 7/Line 231: "...increments in Tc and DM refers..." 

Page 7/Line 237: Substitute attended with attributed 

Page 7/Line 246: "...have shown its effectiveness to reduce..."

Page 7/Line 248: "...acute and prolonged way, perceived soreness..."

References: Please note there are a number of journal titles that need to be checked for capitalization errors.

Author Response

R2.1. The authors attempt to investigate a combination of intermittent pneumatic compression and cold water immersion recovery techniques on various physiological and perceptual measures following basketball and handball matches during a busy competition period. The combined recovery techniques appear to have had beneficial effects on recovery, although these benefits depended on the sex of the athlete.

R2.2. The authors are to be congratulated on their fine study. While this work is promising, there is a major concern with the Introduction of the manuscript and some minor issues with English language and style that will be discussed.

R2.3. Introduction: Of all of the sections of the manuscript, the Introduction did not adequately address the rationale for the study. In essentially one journal page, the authors attempt to address the importance of recovery and some of the methods for optimizing recovery. Yet, nothing is mentioned about using intermittent pneumatic compression. The authors provide a rationale for cold water immersion, but not for IPC (despite using the two in combination). The authors reference other IPC studies in the Discussion, so there are studies that could be included in the Introduction to bolster the strength of that part of the manuscript. The other sections (i.e. Methods, Results, and Discussion) are nicely presented.

- Statement: We thank the reviewer for these affirmative comments. The following information has been added in the introduction to balance the information of both recovery protocols: “On the other hand, compressions garments have been used for exercise-related fatigue recovery. One of these methods is the intermittent pneumatic compression (IPC) pump, which consists of a gradual pressure (0-300 mmHg) gradients applied to facilitate lymph and blood flow [24]. In sports, it has been shown that IPC systems may allow the retained fluid and substances (eg. metabolites) related to muscle damage and neuromuscular fatigue to be removed from the lower and upper limbs [25]. This recovery method are generally designed to simulate the muscle pump, like a typical massage”.

R2.4.Minor issues: The manuscript evidences throughout some minor issues with language, punctuation, capitalization, etc. What follows is a brief list of examples:

Page 1/Line 19: Capitalize May

Page 1/Line 21: Delete is its and replace with it is 

Page 1/Line 26: Replace compared with compare 

Page 2/Line 44: Change thus to these 

Page 2/Line 46: Delete on 

Page 3/Line 104: Rewrite to read ...took part in the study. There were... 

Page 4/Line 132: Do you mean to use the word form instead of from

Page 4/Line 156: I think the TM should be superscript. [Palm Abbe]

Page 6/Line 210: body weight not body Weight 

Page 7/Line 231: "...increments in Tc and DM refers..." 

Page 7/Line 237: Substitute attended with attributed 

Page 7/Line 246: "...have shown its effectiveness to reduce..."

Page 7/Line 248: "...acute and prolonged way, perceived soreness..."

- Statement: Thank you very much for all your comments about the language. All the changes you have proposed have been completed. In addition, an extensive revision of the language has been made.

References: Please note there are a number of journal titles that need to be checked for capitalization errors.

- Statement: Thanks for your contribution. All journal titles have been reviewed and included correctly.

Round 2

Reviewer 2 Report

The authors are to be applauded for significantly improving the Introduction, in particular the information regarding the use of compression as a recovery method. The additional work makes for a stronger manuscript. While the overall strength of the manuscript has been improved, the following minor issues must be addressed before a final editorial decision can be made:

Line 51: "...shown that these types of sports also cause mental..."

Line 58: Delete comma after exposed 

Line 59: Delete without and comma after recovering 

Line 70: Compression not compressions 

Line 76: Do you mean simulate or stimulate?

Line 82: Add and to therefore (i.e. and therefore, ...)

Line 82: "...the recovery of these athletes

Line 94: "...b) compare

Line 114: "...championship, which was part of..."

Line 128: time of contraction 

Line 130: followed previous reported protocols...

Line 152: Players self-reported sleeping hours.

Line 169: intermittent pneumatic compression

Line 177: I think you mean to say seated, not sited

Line 183: *Data was presented as..."

Line 188: "...set at p<0.05."

Table 1: Missing SD for Tc PreMatches

Table 1 and other tables: For consistency, please make sure all values go to 2 decimal places for means and standard deviations

Line 249: Indent issue

Line 251: "...both sexes had a higher value of TQR and a lower value of DOMS."

Line 262: "...fixture periods that are reduced,..."

Limitations is listed as #6 and Conclusions listed as #5. Also, spacing of Limitations needs to be addressed.

References still need to be addressed. There are issues with missing reference information (e.g., #3, #14, #23, #38, #39, #43, #55 missing page information), journal formatting (e.g., #7 Medicine, #25, #32, #34, #49, #57 and vs. &), capitalization (e.g., #9, #33, #43, #44, #49, #50, #55, #57), and capitalization after : (e.g., #5, #11, #17, #18, #20, #24, #31, #47). A thorough overview is recommended.

Author Response

We really appreciate all your recommendations and opportunity to clarify some issues. According to the references, we have solved some of the problems you mentioned to us and we have also abbreviated the title of the journals that were not abbreviated previously. Thank you for all. Please find the corrections in red color inside the document.

Line 51: "...shown that these types of sports also cause mental..."

- Thanks for your comment. This has been changed.

Line 58: Delete comma after exposed 

- Thanks for your comment. This has been deleted.

Line 59: Delete without and comma after recovering 

- Thanks for your comment. This has been deleted.

Line 70: Compression not compressions 

- Thanks for your comment. This has been changed.

Line 76: Do you mean simulate or stimulate?

- Thanks for your comment. We certainly wanted to mean stimulate.

Line 82: Add and to therefore (i.e. and therefore, ...)

- Thanks for your comment. This has been added.

Line 82: "...the recovery of these athletes

- Thanks for your comment. This has been changed.

Line 94: "...b) compare

- Thanks for your comment. This has been changed.

Line 114: "...championship, which was part of..."

- Thanks for your comment. This has been changed.

Line 128: time of contraction 

- Thanks for your comment. This has been changed.

Line 130: followed previous reported protocols...

- Thanks for your comment. This has been changed.

Line 152: Players self-reported sleeping hours.

- Thanks for your comment. This has been changed.

Line 169: intermittent pneumatic compression

- Thanks for your comment. This has been changed.

Line 177: I think you mean to say seated, not sited

- Thanks for your comment. We certainly wanted to mean seated.

Line 183: *Data was presented as..."

- Thanks for your comment. This has been changed.

Line 188: "...set at p<0.05."

- Thanks for your comment. This has been changed.

Table 1: Missing SD for Tc PreMatches

- Thanks for your comment. This has been added.

Table 1 and other tables: For consistency, please make sure all values go to 2 decimal places for means and standard deviations

- Thanks for your comment. This has been revised.

Line 249: Indent issue

- Thanks for your comment. This has been modified.

Line 251: "...both sexes had a higher value of TQR and a lower value of DOMS."

- Thanks for your comment. This has been amended.

Line 262: "...fixture periods that are reduced..."

- Thanks for your comment. This has been amended.

Limitations is listed as #6 and Conclusions listed as #5. Also, spacing of Limitations needs to be addressed.

- Thanks for your comment. This has been replaced.

References still need to be addressed. There are issues with missing reference information (e.g., #3, #14, #23, #38, #39, #43, #55 missing page information), journal formatting (e.g., #7 Medicine, #25, #32, #34, #49, #57 and vs. &), capitalization (e.g., #9, #33, #43, #44, #49, #50, #55, #57), and capitalization after : (e.g., #5, #11, #17, #18, #20, #24, #31, #47). A thorough overview is recommended.

- Thanks for your comment. References has been revised and amended.